# The Spread of Lone Star Ticks (*Amblyomma americanum*) and Persistence of Blacklegged Ticks (*Ixodes scapularis*) on a Coastal Island in Massachusetts, USA

**DOI:** 10.3390/insects15090709

**Published:** 2024-09-17

**Authors:** Richard W. Johnson, Patrick Roden-Reynolds, Allison A. Snow, Stephen M. Rich

**Affiliations:** 1Martha’s Vineyard Tick-Borne Illness Reduction Initiative, Edgartown, MA 02539, USA; mvticks@gmail.com (R.W.J.); biologist@dukescounty.org (P.R.-R.); 2Department of Evolution, Ecology, and Organismal Biology, Ohio State University, Columbus, OH 43210, USA; snow.1@osu.edu; 3Department of Biology, University of Massachusetts, Amherst, MA 01003, USA; 4Laboratory of Medical Zoology, Department of Microbiology, University of Massachusetts, Amherst, MA 01003, USA

**Keywords:** lone star tick, blacklegged tick, range expansion, yard survey, active tick surveillance, tick density, tick-borne illness

## Abstract

**Simple Summary:**

When a species expands its geographic range, it may displace similar species in the new region. In the northeastern USA, lone star ticks (*Amblyomma americanum*) have encroached on the range of blacklegged ticks (*Ixodes scapularis*), becoming more numerous in a wide range of habitats. Both species carry human disease agents, and both rely heavily on white-tailed deer (*Odocoileus virginianus*) for reproduction. We used 1265 yard surveys conducted in 2011–2024 to document the timing and pattern of lone star tick establishment on the island of Martha’s Vineyard (MV), Massachusetts. Increases in lone star ticks coincided with new cases of ehrlichiosis and reports of the alpha gal syndrome “red meat” allergy. To provide an index of current tick abundance, we used drag sampling to quantify the numbers of each tick species per kilometer of trail at 14 study sites on eastern MV where both species have coexisted for a decade. Lone star ticks are now ubiquitous at the study sites, while blacklegged ticks are persisting at relatively high levels in the woods. We conclude that the risk of human exposure to pathogens carried by blacklegged ticks remains high and is complemented by additional health risks associated with lone star ticks.

**Abstract:**

In the northeastern USA, the distribution of lone star ticks (*Amblyomma americanum*) has expanded northward in recent decades, overlapping with the range of blacklegged ticks (*Ixodes scapularis*). Blacklegged ticks carry pathogens for diseases such as Lyme, babesiosis, and anaplasmosis, while bites from lone star ticks cause other diseases and the alpha-gal syndrome allergy. Lone star ticks can become so abundant that they are perceived as more of a public health threat than blacklegged ticks. Using the island of Martha’s Vineyard, Massachusetts, as a case study, we analyzed data from a total of 1265 yard surveys from 2011 to 2024 to document lone star tick presence and subsequent expansion from two peripheral areas, Chappaquiddick and Aquinnah, to all six towns. The timing of lone star tick expansion on Martha’s Vineyard closely matched an increase in tick submissions to a pathogen testing center. At Chappaquiddick, drag sampling carried out in June 2023 and 2024 showed that both tick species were most common at wooded sites, where blacklegged nymphs were somewhat more abundant than lone star nymphs. However, lone star ticks occurred in a wider range of natural and peridomestic habitats than blacklegged nymphs, making them far more challenging for people to avoid and manage.

## 1. Introduction

Blacklegged ticks (*Ixodes scapularis*, also known as deer ticks) carry pathogens causing Lyme disease, babesiosis, anaplasmosis, and other illnesses, and depend on white-tailed deer (*Odocoileus virginianus*) for their adult blood meal and reproduction [1]. In the northeastern USA, endemic levels of tick-borne diseases have increased dramatically over the past half-century, in concert with the growing prevalence of white-tailed deer and blacklegged ticks [2,3,4]. After much of the New England landscape was cleared for agriculture in the mid-1900s, reforestation and forest fragmentation around cities and towns led to large increases in deer abundance [3,4]. The lack of predators and hunting pressure has allowed deer to move into suburban and urban green spaces, further exacerbating the risk of human exposure to blacklegged ticks [5]. Lyme disease is now the most common vector-borne disease in the USA, representing a major focus for public health workers, physicians, and government agencies [6].

Another deer-dependent tick species, *Amblyomma americanum*, known as the lone star tick, has recently extended its geographic range along the eastern seaboard, overlapping with the range of blacklegged ticks to become the dominant species where these species co-occur [7,8]. Active and passive surveillance studies have documented lone star range expansion in New Jersey, New York, Connecticut, Rhode Island, and Massachusetts, often beginning along coastal areas before extending further inland [7,9,10,11]. Lone star ticks also occur in the southern USA and are sometimes referred to as turkey ticks because they feed on wild turkey (*Meleagris gallopavo*), as well as other species [12]. However, white-tailed deer are considered the major blood meal host for all three life stages of lone star ticks, which are vectors for several diseases and are linked to the alpha-gal syndrome allergy [11,13,14]. Although climate change is often invoked as contributing to the northward range expansion of lone star ticks, Rochlin et al. [3] noted that the near-eradication of deer populations prior to the mid-1900s followed by their recovery in re-forested landscapes also has played a role in allowing lone star ticks to re-colonize their former range in southern New England.

Several studies have shown that lone star ticks far outnumber blacklegged ticks where their ranges overlap [7]. However, little is known about whether the population density of blacklegged ticks declines after the establishment of lone star ticks [5]. Detecting sustained declines in tick densities is challenging due to extensive spatiotemporal variability in their abundance [15,16,17,18], but such data are needed to assess the risk of exposure to tick-borne illnesses. By studying tick densities at the leading edge of the lone star tick’s northern range expansion, we can surveil whether densities of blacklegged ticks are altered as lone star ticks become established. Changes in density could be caused by competition for blood meal hosts. Alternatively, deer may acquire resistance to blacklegged ticks following lone star tick feeding [19]. It is also possible that these two tick species have no effect on each other’s population levels, despite differences in their relative densities post-invasion. Changes in their population levels could be due to a variety of other factors, such as extreme heat, cold, drought, host densities, and/or differential susceptibility to entomopathogenic agents [20,21,22,23].

For the present study, we chose the island community of Martha’s Vineyard, Massachusetts, where blacklegged ticks have been endemic for several decades, while lone star ticks have only recently emerged. To examine interspecies interaction, we employ two types of data: (1) long-term incidence reports from active surveillance of residential properties to document arrival and spread of lone star ticks and (2) recent estimates of density of these species. Data from yard surveys conducted in 2011–2024 show that lone star distributions began at the eastern and western ends of the island around 2014–15 and then spread more widely. On the eastern peninsula/island of Chappaquiddick, we used drag sampling to compare lone star and blacklegged tick densities in wooded vs. open areas in 2023 and 2024. In wooded areas, where both species were present, blacklegged nymphs outnumbered lone star nymphs. We conclude that at present, blacklegged ticks are persisting at levels that clearly merit continued public health outreach and intervention, while lone star ticks are ubiquitous in a wider range of habitats, adding to the public health burden of tick-related ailments on Martha’s Vineyard.

## 2. Materials and Methods

### 2.1. Location

The island of Martha’s Vineyard (MV) is 250 km^2^ large and located 8 km south of Cape Cod, MA. The island comprises six towns—Aquinnah, Chilmark, West Tisbury, Tisbury, Oak Bluffs, and Edgartown (Figure 1). For this study, we conducted yard surveys in all six towns to document the presence/absence of each tick species over time. We also recorded tick densities on Chappaquiddick Island to examine the persistence of blacklegged ticks after lone star ticks had invaded. MV has a year-round population of ~20,800 people, with a large influx of seasonal residents and visitors in summer. Chappaquiddick Island (16 km^2^) is part of the town of Edgartown and is usually connected to the main island by a barrier beach. Chappaquiddick has several hundred seasonal homes plus a small year-round population. Geologically, MV originated as part of a terminal glacial moraine and associated outwash near the end of the Wisconsin Glaciation [24]. Higher elevation morainal deposits support mesic deciduous forests, while lower outwash plains are dominated by oak–pine forests and ericaceous shrubs typical of sandy soils [24]. MV has a high deer density, ranging from a minimum of 19 deer per km^2^ (50 deer per square mile) in areas open to hunting to substantially higher deer densities in areas closed to hunting [25].

### 2.2. Yard Surveys

Although blacklegged nymphs are most common in woods, many people are bitten in peridomestic areas around their homes [26,27]. We used data from 1265 yard surveys of 827 unique properties to document spatial and temporal changes in the occurrence of blacklegged (BL) and lone star (LS) ticks from 2011 to 2024. This effort was conducted as part of the Martha’s Vineyard Tick Program [28]. Each year, surveys were carried out between 15 May and 31 July to include the seasonal peak abundance of BL nymphs, LS nymphs, and LS adults on MV. Most cases of Lyme disease occur in the summer months due to bites from BL nymphs [26,29], hence our focus on this life stage, whereas BL adults are mainly encountered in the fall and spring.

In 2011–2015, we offered tick surveys to property owners on Chappaquiddick Island and in the town of Chilmark. From 2016 onward, surveys were conducted in all six towns (including Chappaquiddick) in response to requests from property owners who learned about this program from our outreach activities. Each property was visited at least once during a summer, and some were visited multiple times between 2011 and 2024.

In each surveyed yard, we used a hybrid drag-and-flag method of sweeping with a 1 m^2^ white cloth attached to a wooden handle. We swept the cloth over and around areas where ticks were most likely to be found, including the edge of the yard, shaded areas, gardens, brush piles, and other potential tick habitats. Many yards were surrounded by natural vegetation. We also dragged for ticks in mowed lawns around buildings. Ticks were collected, identified, and counted. We did not record the life stage of the collected ticks for this study. Also, because the primary purpose of the surveys was to educate and inform the property owners, we did not quantify the number of ticks collected per unit of time or distance.

To document the timing and distribution of LS tick establishment across the island, we mapped data from the yard surveys by using ArcGIS Pro 3.0.3. Town and parcel data were sourced from the Dukes County Data Hub hosted by The Martha’s Vineyard Commission. Yard survey data for the maps were pooled for these intervals: 2011–15, 2016–19, and 2020–24. If the same property was surveyed more than once during each interval, we used the average of 2 or more surveys. For two areas, Chappaquiddick Island and the town of Chilmark, sample sizes were large enough to compare the proportion of yards where we found at least 3 ticks per species during the same three intervals (2011–15, 2016–19, and 2020–24). These data were of interest because LS establishment occurred much earlier on Chappaquiddick than in Chilmark.

### 2.3. Passive Surveillance of Lone Star Tick Expansion

To further investigate the timing of LS tick expansion on Martha’s Vineyard, we queried the public database of the crowd-funded TickReport^TM^ testing center [30]. TickReport was developed at the Laboratory of Medical Zoology, University of Massachusetts, Amherst, in 2006. Since 2020, this service has been provided by MedZu, Inc. (Amherst, MA, USA), under a licensing agreement with University of Massachusetts. For a small fee, members of the public can send ticks to TickReport for identification and pathogen testing. This service has been available consistently since 2006, with the exception of mandatory furloughs due to COVID-19 for 19 days in June 2020 and the months of December 2020–late April 2021. Here, we report submissions of 742 LS nymphs and adults found on human hosts in Massachusetts from 2012 to 2023, 337 of which were encountered on MV.

### 2.4. Tick Densities along Trails on Chappaquiddick

To quantify the current densities of BL and LS ticks on Chappaquiddick, drag sampling was carried out along hiking trails from 31 May to 23 June 2023 and from 24 May to 21 June 2024. Lone star ticks are known to occur in a wider range of habitats than BL ticks [31], so we compared tick densities in open grasslands vs. shady woods. We selected a total of 5 open sites, which we refer to as meadows, and 9 wooded sites (Table 1, Figure 2). All 5 open sites were sampled in both years, but 2 of the 5 wooded sites that were used in 2023 could not be re-sampled in 2024 due to removal of leaf litter on the trail. Therefore, four new wooded sites were added in 2024 to obtain better coverage of this habitat (Table 1). To provide a shade index and information about tree species composition at each wooded site, we recorded the presence/absence of tree branches over the center of the trail and identified these tree species at 30–50 equally spaced observation points along each trail (Appendix A). At observation points where no overarching branches were present, we recorded the presence/absence of high shrubs (>1 m tall), which also cast shade along the trail.

The distance over which sampling was performed ranged from 139 to 763 m per site (Table 1), depending on the local trail system. Shrub thickets and a dense shrub understory in most wooded areas precluded the use of replicated sampling within multiple plots, as used in other studies (e.g., [32,33,34]). At each site, ticks were sampled by dragging a 1 m^2^ piece of white cloth along the edges of hiking trails, as in Snow et al., 2023 [18]. The cloth was checked every 12 m to remove and count ticks. When lone star larvae were encountered, a lint roller was used to remove larvae between drag sweeps, and we noted the number of sweeps (12 m long) with clusters of at least 50 LS larvae. At 3 open sites, we dragged cloth along both sides of the 1.9 m wide trail to obtain the sampling distances shown in Table 1.

We drag-sampled each site five days per year, with at least two days between visits to the same site, and we alternated the order and times of day when each site was sampled. At Slater Woods, only four sample dates were used because managers had cleared leaf litter away from the trail. Previous studies have shown that LS ticks often quest during drier periods of the day than BL ticks [35]. However, we did not observe obvious differences between morning and afternoon densities of either tick species, nor did we find more ticks following longer intervals (e.g., 5 days) between visits to the same site. All field work was conducted by the same person (A. Snow). The densities of each tick species and life stage are reported as the mean number of ticks per 0.5 km of trail (N = 5 days) to allow for comparisons across sites and years. For wooded sites, we used paired t-tests to analyze the mean densities of BL vs. LS nymphs. To compare tick densities in wooded vs. open sites, we calculated the mean number per 0.5 km for five days per site, averaged across all sites in the group for each year.

## 3. Results

### 3.1. Yard Surveys

After averaging tick numbers from properties with multiple surveys for a given time frame, we report data from 189 unique properties for 2011–2015, 419 for 2016–2019, and 339 for 2020–2024 (Figure 1). In all years, we did not survey for ticks in the Manuel F. Correllus State Forest or MV Airport; both are located in the center of MV and lack residential properties.

Very few BL adults were encountered, and the majority of LS ticks were nymphs, while adults were also common. Dog ticks (*Dermacentor variabilis*) were counted, but these data are not included in the present study. We also collected a few adult and nymph Asian longhorn ticks (*Haemaphysalis longicornis*), which have recently appeared on MV.

Blacklegged nymphs were common during all three time periods, while LS ticks were rare or absent initially (Figure 1). LS ticks had been detected on several scattered properties on Chappaquiddick by 2015 but were still rare in Chilmark (one property had a single LS tick). By 2019, at least one LS tick was collected in all six towns, but only one survey in both Oak Bluffs and Tisbury had LS ticks, with one and two ticks collected, respectively. By 2024, multiple yards in all six towns had three or more LS ticks. LS tick abundance has increased across northern regions of MV and has remained high in Chappaquiddick and Aquinnah.

Sample sizes for Chappaquiddick and the town of Chilmark were large enough to examine how the proportion of yards with at least three BL or LS ticks changed over time (Figure 3). In 2011–2015, 8% and 0% of yard surveys had at least three LS ticks for Chappaquiddick and Chilmark, respectively. In 2020–2024, their abundance increased to 85% of yards on Chappaquiddick and 65% for Chilmark. The proportion of yards with BL ticks was variable. Initially, in 2011–2015, the frequency of yards with at least three BL ticks in Chilmark was 73%, compared with only 33% on Chappaquiddick. On Chappaquiddick, BL ticks gradually became more common, increasing from 33% to 50% in 2020–2024. However, in Chilmark, BL frequencies declined from 82% of surveyed yards in 2016–2019 to 55% in 2020–2024.

### 3.2. Passive Surveillance of Lone Star Tick Expansion

From 2012 to 2023, annual submissions of LS ticks to TickReport for identification and pathogen testing reflected the increases that we documented in the yard surveys. Submissions for the state of Massachusetts were negligible until 2014, when 19 ticks were received (Figure 4). Submissions from MV started in 2016, with 5 submissions, and increased to 83 in 2023. For the state-wide total of 742 LS ticks submitted from 2012 to 2023, 81% were from Cape Cod and nearby islands, and 45% (337) were from Martha’s Vineyard.

### 3.3. Tick Densities along Trails on Chappaquiddick

#### 3.3.1. Wooded Sites

Both tick species were much more common in the wooded sites than in the open sites (Figure 5). Trails in the wooded sites were shaded throughout, typically with shrubs such as black huckleberry (*Gaylussacia baccata*), arrowwood (*Viburnum dentata*), and hazelnut (*Corylus americana*) in the understory. Pitch pine (*Pinus rigida*) was common at four of the sites, including Packard Woods and Mytoi Woods, and oaks (*Quercus* spp.) were common at all sites (Appendix A).

At the wooded sites, the densities of BL nymphs per 0.5 km averaged 63 in 2023 and 68 in 2024, while those of LS nymphs averaged 33 and 51, respectively (Figure 5, Table 2, Appendix A). At each wooded site and year, we collected more BL nymphs than LS nymphs, and the average ratios of BL nymphs to LS nymphs were 2.1:1.0 in 2023 and 1.4:1.0 in 2024. Paired t-tests showed that the density of BL nymphs was significantly greater than that of LS nymphs each year (*p* < 0.01).

Very few BL adults were encountered (Appendix A), while LS adults were common, with means of 11 and 32 adults per 0.5 km in 2023 and 2024, respectively. Clusters of ≥50 LS larvae were found at two of the three sites that were sampled in both years, Packard Woods and Mytoi Woods, and were more common at Packard Woods in 2024 compared with 2023 (mean of five vs. one larval cluster per 0.5 km). In 2024, we also found LS larval clusters at Sampson Woods and Pocha Woods (Appendix A).

#### 3.3.2. Open Sites

Very few BL ticks were found at the five open sites, all of which were sampled in both years (Figure 5 and Figure 6). The densities of LS nymphs per 0.5 km averaged 11 and 14 across sites in 2023 and 2024, respectively, while those of LS adults averaged 4 and 6 per 0.5 km each year. The average LS densities varied across sites and were as great as 20–25 LS nymphs per 0.5 km at Brine’s Meadow and Handy Meadow (Figure 6). In both years, the greatest density of LS adults was seen at Wasque Meadow, averaging 9 vs. 11 LS adults per 0.5 km in 2023 vs. 2024, respectively. Clusters of LS larvae were not found at any of the five open sites.

## 4. Discussion

### 4.1. Establishment and Spread of Lone Star Ticks

In the northeastern USA, LS ticks have been expanding their range northward along the coast and patterns of colonization have been uneven, often emanating from early “hot spots” found on islands and peninsulas. Local examples of initially isolated populations of LS ticks include Manresa Island, CT; Prudence Island, RI; and Tuckernuck Island, MA [7,18,37]. On Martha’s Vineyard, the presence of LS ticks was sporadic at first. Lone star ticks were seen occasionally as early as 1985 and were breeding on Cape Pogue, Chappaquiddick, and in Aquinnah by 2015 [38,39]. Our yard surveys show that a surge in the abundance of LS ticks on MV did not take place until ~2016–19, occurring first on the eastern island/peninsula of Chappaquiddick and the western peninsula of Aquinnah, before spreading to the rest of the island during the past decade. In 2020–2024, we found LS ticks in 85% of the yards surveyed on Chappaquiddick. It seems likely that similar patterns of spread from initial hot spots to adjacent areas are occurring in other areas along the coast of New England. LS ticks also have been documented in urban yards, such as those on Staten Island, NY [5].

Stafford et al., 2018 [7] reviewed early occurrences of LS ticks in New England that paved the way for further spread. Presumably, initial establishment varies due to episodic dispersal by birds [40]. The persistence and continued spread of LS ticks depends on the availability and movements of blood meal hosts such as deer and wild turkey, the investigation of which was not part of our study. Lone star adults find mates while feeding on deer, and their ability to do so may be lower during the early stages of establishment when tick densities are low. Further research is needed to test this hypothesis. As with other invading species, biotic and abiotic conditions that favor LS tick establishment are likely to vary in time and space, resulting in a lag period before population growth becomes more apparent [41,42].

By the time a new tick species becomes common enough to engender public awareness, further evidence of range expansion may be seen in the numbers of ticks that residents submit to laboratories for identification and pathogen testing. For Martha’s Vineyard, LS tick submissions to TickReport began in 2016 and continued steadily thereafter (Figure 4). Farther south, in Monmouth County, New Jersey, the number of LS ticks that were submitted for pathogen testing increased about five-fold in the decade after 2006, while submissions of BL ticks remained level [10,11]. During this period, LS ticks were spreading from the coast to more central areas of New Jersey [11]. In Connecticut, submissions of LS ticks showed a 3.5-fold increase between 1996 and 2016, mostly concentrated in the southwestern coastal area of Fairfield County, where Manresa Island is located [7]. In Massachusetts and elsewhere, further expansion of LS tick distribution can be investigated by monitoring submissions to tick-testing laboratories.

### 4.2. Frequencies of Blacklegged Ticks in Yards

Yard surveys showed that endemic BL ticks are common across Martha’s Vineyard and are most frequently encountered in areas with morainal topography and mesic deciduous forests, including the towns of Chilmark and West Tisbury (Figure 1). In 2012–2019, BL tick occurrence in yard surveys was lower on Chappaquiddick (~33–50%) than in Chilmark (73–82%; Figure 3), perhaps due to sandier soils [24] and drier microsite conditions for nymphs and small mammals on much of Chappaquiddick. In 2020–2024, BL tick occurrence in Chilmark yards declined to ~55%. After completing each yard survey, we met with the property owner to discuss our findings and potential steps to reduce the number of ticks in their yard and to avoid getting ticks on themselves and their pets. At least part of the BL tick decline in Chilmark may be related to preventative measures that homeowners implemented after having tick surveys completed and discussing options to reduce risk, such as using acaricide applications and tick-preventing landscaping methods. Because BL ticks depend on leaf litter and shady environments to avoid desiccation, as well as small mammals for juveniles’ blood-meal hosts, strategic management of peridomestic areas can be used to reduce their abundance [43]. In contrast, LS ticks are hardier and more frequently encountered in unshaded, open habitats [31], making them more difficult to manage with similar strategies.

### 4.3. Densities of Lone Star and Blacklegged Ticks

LS ticks were common, and BL ticks were rare in open meadows, as expected based on previous studies of habitat preferences for each species [31]. Both species were more common at wooded sites, where deer were frequently encountered. Densities of LS nymphs were lower than those of BL nymphs at wooded sites (Figure 3). In contrast, other researchers have reported that LS nymphs and adults greatly outnumber BL nymphs in wooded habitats [7]. In deciduous woods on Long Island, NY, LS nymphs became five times more abundant than BL nymphs between 1994 and 1997 [44]. Likewise, in Monmouth County, New Jersey, where both species were monitored by drag sampling at forest study sites, Jordan et al., 2022 [45] reported that nymphal densities of lone star ticks were ~three–five times greater than those of blacklegged ticks. They documented declines in densities of blacklegged nymphs from 2017 to 2021, while lone star densities remained steady.

Due to the strong predominance of LS ticks compared with BL ticks in other studies, some researchers have speculated that BL tick populations may decline after LS establishment [5,46]. However, determining whether consistent declines have occurred is challenging, partly because density data from drag-sampling studies are notoriously variable across days, sites, and years, and large-scale sampling efforts are needed for statistically rigorous comparisons [15,16,17]. Another challenge is that densities of BL nymphs may be affected by year-to-year fluctuations in populations of their small mammal hosts, such as white-footed mice (*Peromyscus leucopus*), in which case an observed decline in BL ticks may turn out to be temporary [5,32].

Here, we report average densities of 63 and 68 BL nymphs per 0.5 km, based on five–seven wooded sites per year (Figure 5, Table 2, Appendix A). These values represent a rough estimate of current tick densities on Chappaquiddick, but variation among sites and years precludes us from considering this to be a precise baseline for comparisons with other locations or the same location over time. With this caveat in mind, we note that the average densities of BL nymphs on Chappaquiddick are comparable to those from our previous surveys on nearby Nantucket Island (46–67 nymphs/0.5 km) which took place prior to LS establishment (Table 2; [18]). Taken together, these comparisons suggest that BL nymphs on Chappaquiddick occur at relatively high densities after LS ticks became ubiquitous.

### 4.4. Implications for Public Health

In regions where LS ticks are abundant, residents are likely to encounter them more frequently than BL ticks, which prefer shadier habitats and are stationary when seeking a blood meal. In contrast, LS ticks move quickly toward their hosts and are able to traverse pavement and other barriers while questing. LS ticks may be better adapted to survive in drier, hotter habitats, such as mowed lawns, than BL ticks, increasing the likelihood of encountering LS ticks in yards even when maintained free of overgrowth. Future studies could aim to quantify abundance or densities of LS ticks in interior portions of mowed lawns several meters away from brush or leaf litter edge. On Martha’s Vineyard, LS ticks are frequently encountered in peridomestic areas during May–July (Figure 1) and also in August, when we have observed that larval “tick clusters” are most prevalent. All life stages of LS ticks represent a pervasive nuisance associated with irritating bites, causing some people to avoid outdoor activities and/or cover themselves with tick repellant sprays and insecticide-treated clothing when outdoors. We expect that integrated tick management strategies that are at least somewhat effective for managing BL ticks in residential areas, such as the use of hardscape features and mowed lawns, will be less effective in reducing LS encounters.

Cases of alpha gal syndrome are increasing along with the spread of LS ticks on Martha’s Vineyard, but these data are not reportable by Massachusetts state law, nor are data on southern tick-associated rash illness, heartland virus, or bourbon virus. Martha’s Vineyard clinical providers have reported a substantial increase in diagnosed patients under clinical care for the alpha gal syndrome in recent years, as reported in a local newspaper [47]. Ehrlichiosis is associated with LS ticks and is a reportable illness in Massachusetts. Ehrlichiosis was first documented in Dukes County in 2019, increasing to 3.4 cases per 10,000 people in 2023 (Figure 4B; Dukes County includes Martha’s Vineyard and the small town of Gosnold). All but 2 of the 17 ehrlichiosis cases reported for Cape Cod and the islands from 2018 to 2023 were from residents of Dukes County [37]. Over time, as more people are exposed to LS tick bites, case numbers are expected to increase locally and across the state [10]. Future researchers could examine pathogen prevalence in LS ticks in Massachusetts to gain a better understanding of disease risk.

So far, pathogens carried by BL ticks appear to represent a greater public health threat than LS ticks on Martha’s Vineyard. In 2022–2023, illnesses caused by BL ticks in Dukes County remained heavily weighted towards Lyme disease, followed by less frequent cases of babesiosis and anaplasmosis (with reports of 143.0 cases for Lyme, 7.9 for babesiosis, and 4.1 for anaplasmosis per 10,000 people [48]). Furthermore, it is often noted that Lyme disease cases are routinely underestimated and difficult to monitor [27]. For Martha’s Vineyard, we also note that state-required case reports pertain only to local residents and do not reflect tick-borne illnesses contracted by seasonal visitors.

## 5. Conclusions

To reiterate, the two main goals of this study were to determine how quickly LS ticks have spread across the island of Martha’s Vineyard and how their current densities compare with those of BL ticks post-invasion. Our study contributes to the literature on risks of tick-borne illness in that few previous studies have quantified the frequencies of both LS and BL ticks in residential yards [5]. From the yard surveys, we found that LS ticks became common in two peripheral areas, Chappaquiddick and Aquinnah, by 2016–19. They then dispersed to all towns on the island by 2020–2024. A similar rate of spread of LS ticks among residential properties is likely to occur on the mainland, especially in suburban and rural areas where deer densities are high and there are few impediments to their movements.

Post-invasion, we expect LS ticks to be more common than BL ticks in peridomestic settings due to their wider habitat use, active foraging behavior, and higher fecundity. LS ticks now infest ~85% of residential yards on Chappaquiddick, while BL ticks were found in ~50% of yards (N = 54, 2020–2024). Compared with BL ticks, LS ticks lay more eggs per female (~5000 vs. 3000), have better nymphal survival, are more attracted to CO_2_ emitted by hosts, move more quickly, and travel greater distances when questing [8,31,35,44]. These features make LS ticks much more difficult to avoid and manage than BL ticks.

The second goal of this study was to quantify the densities of LS and BL ticks on Chappaquiddick Island, where they have coexisted during the past decade. Our findings are relevant to speculation about whether BL tick populations may decline after LS ticks have invaded [5,45]. Both tick species were common at the wooded study sites, but unlike previous investigators, we did not find that LS nymphs were much more common than BL nymphs at these sites. Importantly, the density of BL nymphs at wooded sites was relatively high, averaging ~60–70 nymphs per 0.5 km of trail (Table 2; [18]). Therefore, we conclude that current ecological conditions and host–parasite interactions on Chappaquiddick are conducive to the coexistence of both tick species. Whether this is a long-term scenario and whether BL ticks decline elsewhere in the northeastern USA following LS invasion remains to be seen. From a public health perspective, the current burden of tick-borne illness attributed to each tick species appears to be additive on Martha’s Vineyard, calling for continued research, outreach, and preventative measures to reduce people’s exposure to both BL and LS ticks.

## Figures and Tables

**Figure 1 insects-15-00709-f001:**
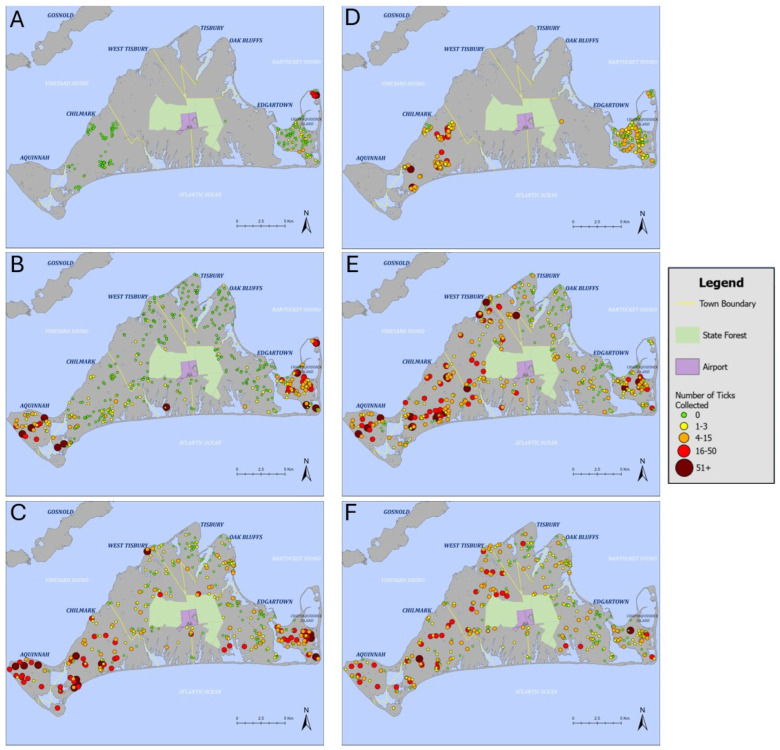
Maps of yard survey data for lone star ticks (*Amblyomma americanum* (**A**–**C**)) and blacklegged ticks (*Ixodes scapularis* (**D**–**F**)). Sample sizes represent numbers of unique sites: top row, 2011–2015, N = 189; middle row, 2016–2019, N = 419; bottom row, 2020–2024, N = 339. Total sample sizes for each town and year are listed in Appendix A.

**Figure 2 insects-15-00709-f002:**
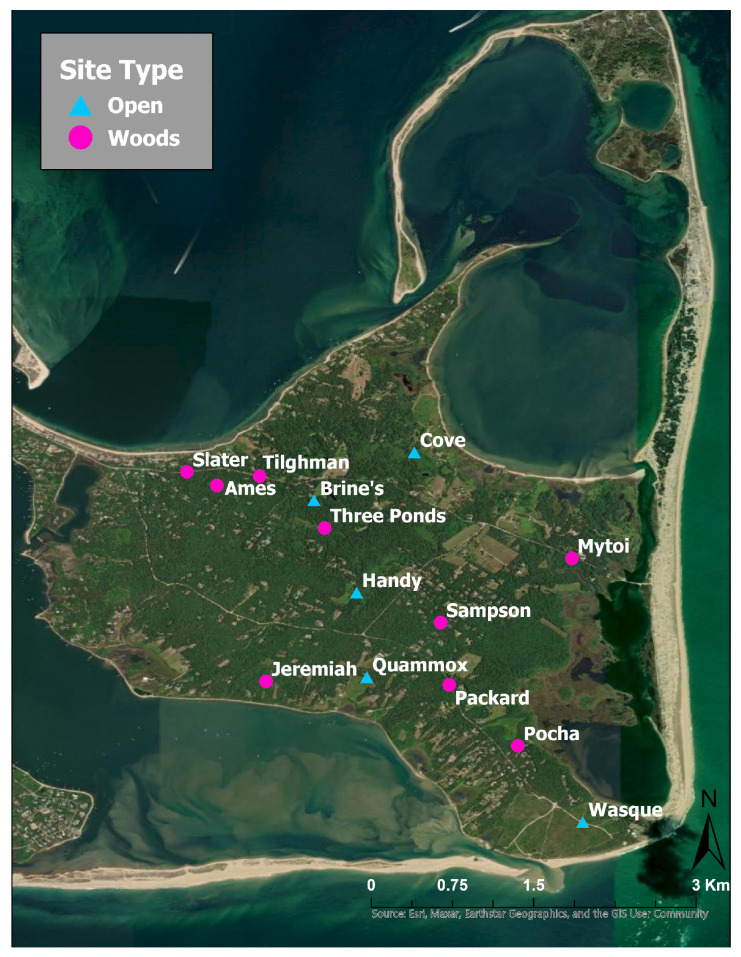
Map of 9 wooded sites and 5 open sites where tick densities per 0.5 km of trail were measured on Chappaquiddick Island, Martha’s Vineyard.

**Figure 3 insects-15-00709-f003:**
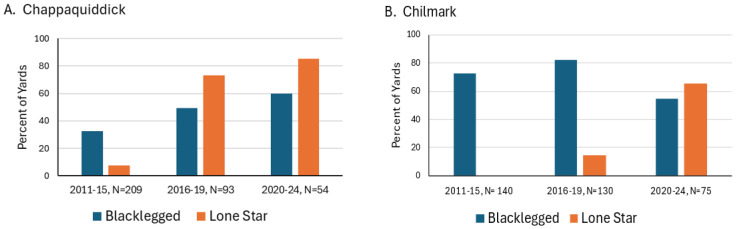
Percent of yards in Chappaquiddick and Chilmark with at least 3 individuals of each tick species, blacklegged ticks (*Ixodes scapularis*) and lone star ticks (*Amblyomma americanum*), in 2011–2024. Sample sizes show the total number of surveys conducted for each period.

**Figure 4 insects-15-00709-f004:**
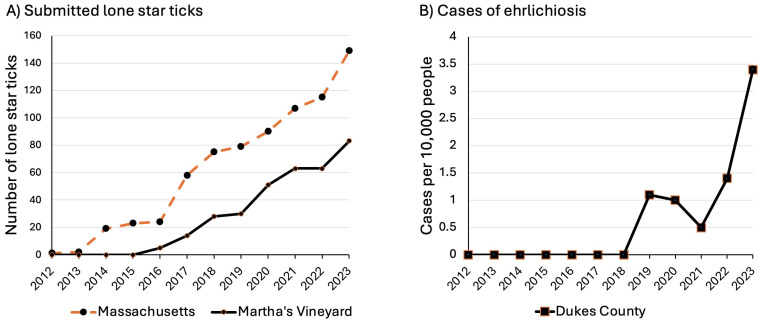
(**A**) Number of lone star ticks (*Amblyomma americanum*) for the state of Massachusetts submitted to TickReport [30] for pathogen testing from 2012 to 2023. Data represent adults and nymphs that had human hosts; N = 742. (**B**) Reported cases of ehrlichiosis for residents of Dukes County (pop. ~20,868), which includes Martha’s Vineyard and the town of Gosnold (pop. < 100) [36].

**Figure 5 insects-15-00709-f005:**
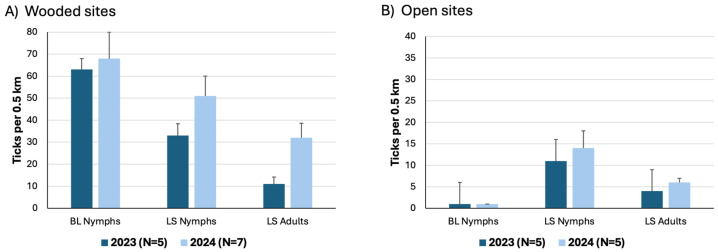
Average densities of each tick species at wooded vs. open sites on Chappaquiddick Island in 2023 and 2024. Based on the average of 5–7 sites in each year; error bars show 1 SE. BL = blacklegged ticks (*Ixodes scapularis*), and LS = lone star ticks (*Amblyomma americanum*). Note difference in vertical scale for Figure 5A vs. Figure 5B.

**Figure 6 insects-15-00709-f006:**
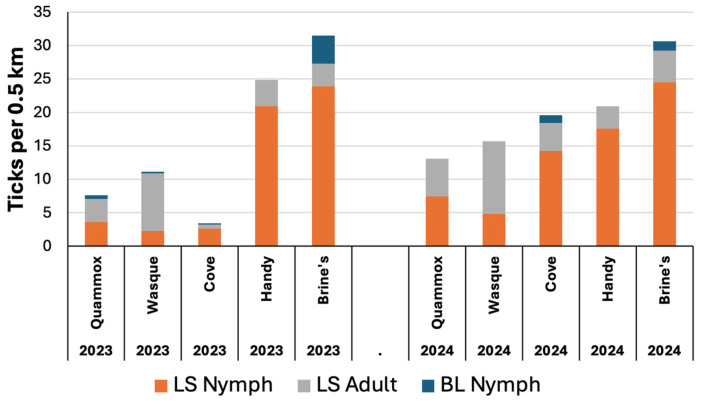
Average tick densities per 0.5 km at open sites in 2023 and 2024. LS = lone star tick (*Amblyomma americanum*), and BL = blacklegged nymphs (*Ixodes scapularis*). N = 5 sample dates per site each year.

**Table 1 insects-15-00709-t001:** Study sites where tick densities were measured on Chappaquiddick Island, Martha’s Vineyard.

Site Name	Site Type	Year	Distance Sampled (m)	GPS Latitude	GPS Longitude
Cove Meadow	Open	2023–24	526	41.38412	−70.47142
Wasque Meadow	Open	2023–24	395	41.35383	−70.45746
Handy Meadow	Open	2023–24	506	41.37263	−70.47621
Brine’s Meadow	Open	2023–24	502	41.38020	−70.47974
Quammox Meadow	Open	2023–24	550	41.36570	−70.47536
Three Ponds Woods	Woods	2023	303	41.37785	−70.47866
Slater Woods	Woods	2023	763	41.38244	−70.49026
Mytoi Woods	Woods	2023–24	553	41.37537	−70.45835
Tilghman Woods	Woods	2023–24	259	41.38209	−70.48424
Packard Woods	Woods	2023–24	638	41.36499	−70.46853
Ames Woods	Woods	2024	205	41.38134	−70.48779
Sampson Woods	Woods	2024	139	41.37010	−70.46924
Jeremiah Woods	Woods	2024	201	41.36530	−70.48372
Pocha Woods	Woods	2024	347	41.36000	−70.46283

**Table 2 insects-15-00709-t002:** Comparison of average densities of blacklegged nymphs (*Ixodes scapularis*) at wooded sites on Chappaquiddick vs. sites on Nantucket. Averages are based on the number of sites each year. Nantucket data are from Snow et al., 2023 [18].

Island	Year	Mean Nymphs per 0.5 km	1 SE	Range	Number of Sites
Chappaquiddick	2023	63	5	44–72	5
	2024	68	12	33–118	7
Nantucket	2020	54	9	35–79	4
	2021	67	16	37–103	4
	2022	46	8	36–70	4

## Data Availability

Data are available from the corresponding author (S.M.R.) upon request.

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
