# Peer review of "The Spread of Lone Star Ticks (Amblyomma americanum) and Persistence of Blacklegged Ticks (Ixodes scapularis) on a Coastal Island in Massachusetts, USA"

_insects, 2024, doi:10.3390/insects15090709_

Round 1
Reviewer 1 Report
Comments and Suggestions for Authors
The manuscript describes results of several different efforts to collect data on the spread of lone star ticks on the island of Martha's Vineyard. The manuscript is well-written and interesting. Some of the data presented were collected as part of earlier outreach efforts, which brings some limitations in terms of study design. There is also a lot of redundant and overly detailed explanations. Finally, there is a lack of appropriate statistical analysis which is needed.
Specific comments:
Methods - This section is quite long and detailed, reading more like a descriptive narrative than a methods section. Some parts (such as discussion of what species were found) belong in the Results section. I suggest making the whole section more concise. The descriptions of tick dragging techniques in particular are overly detailed (this is a standard sampling practice) and repeated in several sections.
The island-wide surveillance portion of the study is interesting but as the authors note, was really an outreach effort rather than a research activity. Sites were not selected at random, sampling effort in terms of time and distance was variable between sites and some sites were sampled more than others. That is fine for a descriptive study of presence/absence, but the lack of precision in data collection makes the use of statistical analyses to compare between towns and years seem slightly misleading. It is also not clear why the authors chose a Fisher’s exact test instead of chi square. I suggest omitting the statistics here and using statistical analyses for the 2023-2024 species comparison in wooded and open areas.
As the first period of the island-wide surveys in 2011 to 2015 only involved sampling in Chilmark and Chappaquiddick, it is difficult to make claims about tick species presence or absence on other parts of the island during that time. While the authors are careful to mention that surveillance area in those first four years was limited to the two towns, it is still a bit confusing to refer to that period as island-wide surveillance or make statements about ticks in parts of the island that were not surveyed.
Table 1 doesn’t need to be in the manuscript – maybe just supplemental materials but include more information on when sampling took place (month?).
For the section on Tick Densities Along Trails on Chappaquiddick, why was there no statistical analysis? This portion of the study is the most important for testing whether the spread of LS has an impact on BL. Therefore, statistical analyses are needed here.
Results - Again, this section could be more concise. It is not appropriate to explain sampling techniques here, as that was covered in the Methods section.
Figure 1: This series of figures is very interesting! It would be helpful to explain that N is the number of unique sites sampled over the time period (it’s in the text but should be in the legend as well). Also, why are the figures numbered in this order if Figure 2 is the first mentioned in the text?
Figure 3: Again, specify what N is. To be honest, I’m not sure what the objective of this figure is. It shows that BL are present in both communities at varying frequency levels, while LS ticks appear later in in Chilmark than in Chappaquidick. That is already clear from Figure 1. I suggest omitting Figure 3.
Figure 4a is good, showing that LS ticks were being sent in from other places in Massachusetts before being sent in from Martha’s Vineyard and also showing that about half the LS ticks submitted came from the island in later years. For Figure 4b, it would be helpful to include all cases of erlichiosis reported in the state over that time period as well as just within the county.
As mentioned earlier, statistical analyses are needed for the results in Figure 5.
Discussion - The discussion section is well-written and makes good points. It could be shorter (avoid reiterating methodology unless you are pointing out a limitation or unique feature of the methods).
Table 2 – it is not necessary to include an actual data table. Just the text of the discussion is sufficient, plus it’s weird to see a table in a discussion section.
Author Response
Reviewer 1
The manuscript describes results of several different efforts to collect data on the spread of lone star ticks on the island of Martha's Vineyard. The manuscript is well-written and interesting. Some of the data presented were collected as part of earlier outreach efforts, which brings some limitations in terms of study design. There is also a lot of redundant and overly detailed explanations. Finally, there is a lack of appropriate statistical analysis which is needed.
REPLY – See detailed responses below.
Specific comments:
- Methods - This section is quite long and detailed, reading more like a descriptive narrative than a methods section. Some parts (such as discussion of what species were found) belong in the Results section. I suggest making the whole section more concise. The descriptions of tick dragging techniques in particular are overly detailed (this is a standard sampling practice) and repeated in several sections.
REPLY – We moved the discussion of which species were found to the Results. Also, we shortened the Methods where possible, keeping in mind that Reviewer 3 asked for more details about our dragging methods (we assume their concern is related to the yard surveys, which were not intended to be quantitative). Some readers will not be familiar with standard sampling methods for ticks, so we retained certain details while deleting repetitive information.
- The island-wide surveillance portion of the study is interesting but as the authors note, was really an outreach effort rather than a research activity. Sites were not selected at random, sampling effort in terms of time and distance was variable between sites and some sites were sampled more than others. That is fine for a descriptive study of presence/absence, but the lack of precision in data collection makes the use of statistical analyses to compare between towns and years seem slightly misleading. It is also not clear why the authors chose a Fisher’s exact test instead of chi square. I suggest omitting the statistics here and using statistical analyses for the 2023-2024 species comparison in wooded and open areas.
REPLY – In the text, we explain that the yard surveys were not designed to be quantitative. The reviewer raises a good point about not using statistical analyses to make comparisons between towns and years due to unequal sampling efforts. Therefore, we have deleted these statistical comparisons.
The reviewer also requested that we use statistics to compare species composition in wooded and open areas. This comment is addressed below, Comment 5.
- As the first period of the island-wide surveys in 2011 to 2015 only involved sampling in Chilmark and Chappaquiddick, it is difficult to make claims about tick species presence or absence on other parts of the island during that time. While the authors are careful to mention that surveillance area in those first four years was limited to the two towns, it is still a bit confusing to refer to that period as island-wide surveillance or make statements about ticks in parts of the island that were not surveyed.
REPLY – To avoid possible confusion, we changed the headings of “Island-wide yard surveys” to simply “Yard surveys” in the Methods and Results. In the text, we are careful to specify which time periods were included in the island-wide surveillance.
- Table 1 doesn’t need to be in the manuscript – maybe just supplemental materials but include more information on when sampling took place (month?).
REPLY – This is a valid point, but one purpose of the table is to show the reader how the number and identity of wooded sites differed in 2023 vs 2024. We retained the table for easier comprehension of this information.
- For the section on Tick Densities Along Trails on Chappaquiddick, why was there no statistical analysis? This portion of the study is the most important for testing whether the spread of LS has an impact on BL. Therefore, statistical analyses are needed here.
REPLY – In the Results, we added statistical tests comparing densities of BL vs LS nymphs at the 9 wooded study sites. Also, we now explain that BL nymphs always outnumbered LS nymphs at each wooded site and year. (This is an index of their relative densities during peak abundance, not a comprehensive comparison of year-round populations, as explained elsewhere.) Difference between open and wooded sites, and between tick species within open sites, are substantial and obvious (Figure 5). We do not think it is necessary to add statistical analysis for these comparisons.
- Results - Again, this section could be more concise. It is not appropriate to explain sampling techniques here, as that was covered in the Methods section.
REPLY – We deleted text in the Results that has been covered in the Methods.
- Figure 1: This series of figures is very interesting! It would be helpful to explain that N is the number of unique sites sampled over the time period (it’s in the text but should be in the legend as well).
REPLY – We now note the number of unique sites in the figure legend.
- Also, why are the figures numbered in this order if Figure 2 is the first mentioned in the text?
REPLY – Figure 1 is mentioned first in the text, in paragraph 2.1 about Location.
- Figure 3: Again, specify what N is. To be honest, I’m not sure what the objective of this figure is. It shows that BL are present in both communities at varying frequency levels, while LS ticks appear later in in Chilmark than in Chappaquiddick. That is already clear from Figure 1. I suggest omitting Figure 3.
REPLY – We explain what N is for this figure. The objective of Figure 3 is to clearly show comparisons between two areas with different histories of lone star invasion vs. BL tick occurrence, and to report quantitative comparisons of presence/absence of LS and BL tick occurrence over time. We do not think that this depth of information is easy for readers to extract from Figure 1, so we have retained this figure.
- Figure 4a is good, showing that LS ticks were being sent in from other places in Massachusetts before being sent in from Martha’s Vineyard and also showing that about half the LS ticks submitted came from the island in later years. For Figure 4b, it would be helpful to include all cases of erlichiosis reported in the state over that time period as well as just within the county.
REPLY – We requested this information from the Massachusetts Dept. of Public Health, but the time needed to obtain it is too lengthy for meeting the journal’s deadline for revisions. Therefore, we were not able to add this to Figure 4b.
- As mentioned earlier, statistical analyses are needed for the results in Figure 5.
REPLY – See Comment 5, above.
- Discussion - The discussion section is well-written and makes good points. It could be shorter (avoid reiterating methodology unless you are pointing out a limitation or unique feature of the methods).
REPLY – We deleted text about methodology that was covered elsewhere.
- Table 2 – it is not necessary to include an actual data table. Just the text of the discussion is sufficient, plus it’s weird to see a table in a discussion section.
REPLY – We moved Table 2 to the Results. This table serves two purposes. First, readers can see average BL nymphs densities across wooded sites and the range of densities that we recorded in 2023 and 2024. Second, it concisely shows how densities on Chappaquiddick compare with those from sites on Nantucket, which lacked LS ticks, in a way that is easier to comprehend than solely writing about it in the text. We view the comparison between Chappaquiddick and our previously published Nantucket study as an important contribution of our paper.
Reviewer 2 Report
Comments and Suggestions for Authors
The objectives of the survey have been achieved. general and specific comments are contained in the attached report summary.

Author Response
Reviewer 2
- Journal relevance – “ticks are not insects”
REPLY – This journal publishes papers about ticks as well as insects. This paper was invited for the special addition on Tick Surveillance and Tick-borne Diseases
- Line 26 – “additional health risks associated with lone star tick should be mentioned in the summary.”
REPLY – The Simple Summary already mentions ehrlichiosis and alpha gal syndrome. We are not able to add more details while staying within the word limit, but readers can find this information in the references cited elsewhere.
- Line 64 - “quantification of the average number of ticks per deer would have added value to this study”
REPLY – Obtaining data on the numbers of ticks per deer was beyond the scope of our study.
- Line 135 – fact sheets about the importance of deer “can be availed to relevant stakeholders”
REPLY – This information is already available for residents of Martha’s Vineyard.
- Lines 244-249 – cases of ehrlichiosis for residents of Massachusetts should be provided if available.
REPLY – We requested this information from the Massachusetts Dept. of Public Health, but the time needed to obtain it is too lengthy for meeting the journal’s deadline for revisions.
Reviewer 3 Report
Comments and Suggestions for Authors
Dear Authors
I have reviewed your manuscript. This study offers valuable insights into the spread and density of Lone Star and Blacklegged ticks on Martha’s Vineyard. Below, I have outlined several comments and suggestions that I believe will strengthen your manuscript.
- Study Scales and Location Selection:
- It would be helpful to introduce the rationale behind selecting two study scales and the criteria for choosing the study areas in the "2.1 Location" section. This will enhance the readers' understanding of the research design and its implications.
- Uniformity of Surveys:
- The tick surveys were not uniformly conducted across all regions and time periods, which might introduce variability and potential biases. I suggest addressing this in the methods or discussion section, explaining how these factors were accounted for in your analysis.
- Detailed Sampling Methodology:
- To improve reproducibility and clarity, I recommend including more details about the drag-sampling method, such as the number of times the cloth was dragged, the total distance covered, and the duration of each session.
- Relevance of Property Owner Interaction:
- While the interaction with property owners is important for community engagement, it may not be directly relevant to the data collection or analysis presented in the study. You might consider relocating this detail to the introduction or discussion to provide context for the community involvement aspect.
- Impact of Sampling Intervals:
- Given that the sampling method involves removing ticks from the environment, the interval between each sampling session could significantly impact the results. Discussing this potential limitation in the discussion section would provide a more comprehensive understanding of the study's findings and their potential biases.
- Lag Period and Host Animals:
- While the discussion on the lag period and host animals' role is important, I understand that your study did not specifically focus on these aspects. A brief acknowledgment of this limitation and a suggestion for future research could add depth to the discussion.
- Public Health Implications:
- The discussion on public health implications is well-presented; however, it’s important to note that the study did not conduct pathogen testing. Framing the discussion related to pathogen-associated illnesses cautiously and distinguishing these observations from the primary findings would enhance the manuscript's scientific rigor.
Overall, your manuscript makes a valuable contribution to the understanding of tick-borne risks on Martha’s Vineyard. I hope these suggestions will help further refine your work.
Best regards,

Author Response
Reviewer 3
Dear Authors
I have reviewed your manuscript. This study offers valuable insights into the spread and density of Lone Star and Blacklegged ticks on Martha’s Vineyard. Below, I have outlined several comments and suggestions that I believe will strengthen your manuscript.
- Study Scales and Location Selection:
- It would be helpful to introduce the rationale behind selecting two study scales and the criteria for choosing the study areas in the "2.1 Location" section. This will enhance the readers' understanding of the research design and its implications.
REPLY - We added a sentence about this in section 2.1 Location. Other criteria for choosing the study areas are explained in the Methods.
- Uniformity of Surveys:
- The tick surveys were not uniformly conducted across all regions and time periods, which might introduce variability and potential biases. I suggest addressing this in the methods or discussion section, explaining how these factors were accounted for in your analysis.
REPLY – At the suggestion of Reviewer 1, we have deleted statistical comparisons between regions (Chappaquiddick vs Chilmark) and years. Figure 1 provides the visual information needed to see the pattern and timing of lone star spread. In the Discussion, we are careful to avoid drawing conclusions that are not well supported by the data shown in Figure 1.
- Detailed Sampling Methodology:
- To improve reproducibility and clarity, I recommend including more details about the drag-sampling method, such as the number of times the cloth was dragged, the total distance covered, and the duration of each session.
REPLY – We assume that this comment is related to the yard surveys. In the text, we
explain that the yard surveys were not intended to be quantitative in terms of the distance or time spent dragging for ticks, which varied among these consultation visits with property owners. Therefore, we do not have any additional information to add.
- Relevance of Property Owner Interaction:
- While the interaction with property owners is important for community engagement, it may not be directly relevant to the data collection or analysis presented in the study. You might consider relocating this detail to the introduction or discussion to provide context for the community involvement aspect.
REPLY – This information has been moved to the Discussion.
- Impact of Sampling Intervals:
- Given that the sampling method involves removing ticks from the environment, the interval between each sampling session could significantly impact the results. Discussing this potential limitation in the discussion section would provide a more comprehensive understanding of the study's findings and their potential biases.
REPLY – In the Methods, we now note that the interval between sampling sessions, which was ~2-5 days, did not appear to affect our results. Our experience and references cited in Snow et al. (2023) suggest that the numbers of ticks removed during sampling is a small fraction of the numbers that can be collected on the same day or a different day. To avoid making the text longer, we decided not to cover this topic in detail.
- Lag Period and Host Animals:
- While the discussion on the lag period and host animals' role is important, I understand that your study did not specifically focus on these aspects. A brief acknowledgment of this limitation and a suggestion for future research could add depth to the discussion.
REPLY – We added a sentence to the Discussion to address this comment.
- Public Health Implications:
- The discussion on public health implications is well-presented; however, it’s important to note that the study did not conduct pathogen testing. Framing the discussion related to pathogen-associated illnesses cautiously and distinguishing these observations from the primary findings would enhance the manuscript's scientific rigor.
REPLY - We added a sentence to the Discussion to address this comment. We assume that readers understand that new data from pathogen testing was not included in our primary findings.